# The Transcriptional Regulator *Prdm1* Is Essential for the Early Development of the Sensory Whisker Follicle and Is Linked to the Beta-Catenin First Dermal Signal

**DOI:** 10.3390/biomedicines10102647

**Published:** 2022-10-20

**Authors:** Pierluigi G. Manti, Fabrice Darbellay, Marion Leleu, Aisling Y. Coughlan, Bernard Moret, Julien Cuennet, Frederic Droux, Magali Stoudmann, Gian-Filippo Mancini, Agnès Hautier, Jessica Sordet-Dessimoz, Stephane D. Vincent, Giuseppe Testa, Giulio Cossu, Yann Barrandon

**Affiliations:** 1Laboratory of Stem Cell Dynamics, School of Life Sciences, Ecole Polytechnique Fédérale Lausanne, 1015 Lausanne, Switzerland; 2Department of Oncology and Hemato-Oncology, University of Milan, Via Santa Sofia 9, 20122 Milan, Italy; 3Department of Experimental Oncology, European Institute of Oncology IRCCS, Via Adamello 16, 20139 Milan, Italy; 4Laboratory of Developmental Genomics, School of Life Sciences, Ecole Polytechnique Fédérale Lausanne, 1015 Lausanne, Switzerland; 5Department of Genetic Medicine and Development, Faculty of Medicine, University of Geneva Medical School, 1211 Geneva, Switzerland; 6BioInformatics Competence Center, UNIL-EPFL, 1015 Lausanne, Switzerland; 7Histology Core Facility, Ecole Polytechnique Fédérale Lausanne, 1015 Lausanne, Switzerland; 8Institut de Génétique et de Biologie Moléculaire et Cellulaire, 67404 Illkirch, France; 9Centre National de la Recherche Scientifique (CNRS), UMR7104, 67404 Illkirch, France; 10Institut National de la Santé et de la Recherche Médicale (INSERM), U1258, 67404 Illkirch, France; 11Université de Strasbourg, 67404 Illkirch, France; 12Division of Cell Matrix Biology and Regenerative Medicine, University of Manchester, Manchester M139PL, UK or; 13Division of Neuroscience, IRCCS San Raffaele Hospital, 20132 Milan, Italy; 14Centre Hospitalier Universitaire Vaudois, 1011 Lausanne, Switzerland; 15Duke-NUS Graduate Medical School, Singapore 169857, Singapore; 16Department of Plastic, Reconstructive and Aesthetic Surgery, Singapore General Hospital, Singapore 169608, Singapore; 17A*STAR Skin Research Labs, Singapore 138648, Singapore

**Keywords:** sensory vibrissae, barrel cortex, *Prdm1*, *Lef1*, Leaf, non-conserved enhancer

## Abstract

*Prdm1* mutant mice are one of the rare mutant strains that do not develop whisker hair follicles while still displaying a pelage. Here, we show that *Prdm1* is expressed at the earliest stage of whisker development in clusters of mesenchymal cells before placode formation. Its conditional knockout in the murine soma leads to the loss of expression of *Bmp2*, *Shh*, *Bmp4*, *Krt17*, *Edar*, and *Gli1*, though leaving the β-catenin-driven first dermal signal intact. Furthermore, we show that *Prdm1* expressing cells not only act as a signaling center but also as a multipotent progenitor population contributing to the several lineages of the adult whisker. We confirm by genetic ablation experiments that the absence of macro vibrissae reverberates on the organization of nerve wiring in the mystacial pads and leads to the reorganization of the barrel cortex. We demonstrate that *Lef1* acts upstream of *Prdm1* and identify a primate-specific deletion of a *Lef1* enhancer named Leaf. This loss may have been significant in the evolutionary process, leading to the progressive defunctionalization and disappearance of vibrissae in primates.

## 1. Introduction

The whisker follicles (or vibrissae—from the Latin vibrio) are complex, self-renewing sensory micro-organs loaded with multipotent epithelial stem cells localized on the snout [1,2]. These stem cells permit the regular replacement of the whisker (a terminally differentiated hair) and, consequently, allow for the continuity of tactile mechanical sensing [3,4,5].

Conserved throughout evolution, whiskers underwent a reduction throughout the primate adaptive radiation [6] and disappeared completely in the human lineage; yet vestiges of whisker capsular skeletal muscles remain in the human upper lip [7]. These specialized hair follicles are bigger both in length and width compared to those of the pelage and are enveloped by vascular sinuses conferring rigidity to the hair shaft. Fibers of striated muscle have an insertion on the capsula and encompass the vascular sinuses. While macro vibrissae are motile and used for distance detecting/object locating, micro vibrissae are immotile and used for object identification.

The processing of whisker-acquired information occurs in the barrel cortex, where each whisker is represented by a discrete and well-defined cytoarchitectonic structure referred to as the barrel [8]. The barrel map occupies a large area of the rodent brain, it is in large part genetically specified and forms early on during development [9]. As the whisker pattern is established earlier and independently from innervation, the hypothesis that whiskers impose their own pattern onto the somatosensory cortex in the homeomorphic fashion has arisen [10].

*Prdm1* (also known as *Blimp1*) is a zinc-finger transcriptional repressor [11] that was shown to be a master regulator controlling terminal differentiation of B-lymphocytes [12,13,14]; it also governs T-cell homeostasis [15] and primordial germ cell specification [16], stem cell maintenance in the sebaceous gland [17], and skin differentiation [18,19]. *Prdm1* was also shown to play a crucial role during whisker development in mice [20].

*Sox2*Cre driven *Prdm1* conditional knockout mice are one of the very rare transgenic animals entirely lacking whisker (vibrissae) follicles, while pelage hair follicles develop physiologically [20]. The loss of this gene impairs whisker development, but the exact stage at which the development is halted has not yet been identified. Furthermore, it is still not known which type of mystacial vibrissae (macro- and/or micro-) are impacted in these mutant mice.

Moreover, Robertson et al. demonstrated that *Prdm1* positive mesenchymal cells give rise to the mature dermal papilla (DP) and expand to form a mesenchymal layer immediately surrounding the hair follicles [20]. However, what this mesenchymal layer gives rise to in the adult whisker is not yet known.

In addition, the reverberation of whisker development halt onto the organization barrel cortex was not investigated yet. The re-organization of the somatosensory cortex is a phenomenon of great evolutionary importance, given the expansion of the brain areas dedicated to the processing of the sensory organs.

In summary, despite its clear role in whisker development, relatively little is known about the regulation of *Prdm1* expression during whisker development. Investigating this subject is of fundamental importance as the loss of regulatory elements in genes might explain the morphogenetic changes that occurred throughout the primate adaptive radiation and that led to the reduction in snout size and vibrissae while hands and eyes of diurnal primates took over as sensory organs.

## 2. Materials and Methods

### 2.1. Mouse Strains

OF1 and C57/B6J2 mice were obtained from Charles River Breeding Laboratories. *Prdm1*MEGFP transgenic mice were kindly provided by Mitinori Saitou; *Lef1*tm1 Rug mice were provided by Rudolf Grosschedl and Werner Held. *Sox2*Cre, *Sox2*CreERT2, *Prdm1*Cre, *Wnt1*Cre, *Prdm1* CA, and ROSAYFP mice were obtained from the Jackson Laboratory (Bar Harbor, United States) (Table 1). The strains carrying the Cre recombinase and ROSAYFP were kept in heterozygosity, while the conditional knockout was performed in homozygosity. Genotyping was conducted using the primers listed in Table 2. All animals were maintained in a 12 h light cycle providing food and water ad libitum. Mice were sacrificed by intra-peritoneal (i.p.) injection of pentobarbital. Experiments were conducted in accordance with the EU Directive (86/609/EEC) for the care and use of laboratory animals and that of the Swiss Confederation. Mating of adult female and male mice was carried out overnight. Time-pregnant mice were sacrificed by injection of pentobarbital and uteri with embryos were removed by dissection.

### 2.2. In Situ Hybridization

RNA-FISH was performed using 8-μm paraffin sections. Digoxigenin-labeled probes for specific transcripts were prepared by PCR with primers designed using published sequences. The mRNA expression patterns were visualized by immunoreactivity with anti-digoxigenin horseradish peroxidase-conjugated Fab-fragments (Roche, Basel, Switzerland), according to the manufacturer’s instructions. The amplification was carried out using the TSA Plus Cyanine 3/5 System (Perkin Elmer, Waltham, United States). *Gli1* and *Wnt10b* ISH were performed with the ACD RNAscope (Newark, United States).

### 2.3. In Vivo Lineage Tracing

For the lineage tracing experiment, *Sox2*CreERT2 crossed with ROSAYFP pregnant females were induced at embryonic day E12 and E12.5 with 2 mg tamoxifen and 1 mg of progesterone (Sigma-Aldrich, St. Louis, United States) by intraperitoneal injection. The transgenic animals were retrieved at E17 and perinatally and then processed for histology and immunostaining.

### 2.4. Proliferation Experiments

Pregnant female mice carrying *Prdm1*MEGFP embryos were injected with 200 µL of EdU (30 mg/ mL) and analyzed 2 h after the first injection. Embryos were retrieved, genotyped under the UV lamp, and processed for histology and immunostaining.

### 2.5. Histology and Immunostaining

All samples were removed and fixed overnight in 4% paraformaldehyde at 4 °C. Tissues were washed three times in PBS for 5 min and incubated overnight in 30% sucrose in PBS at 4 °C; eventually, they were then embedded in OCT and kept at −80  °C. Sections of 10 µm thickness were cut using a CM3050S Leica cryostat (Leica Microsystems, Wetzlar, Germany).

Sections were incubated in blocking buffer (1% BSA, 0.3% Triton in PBS) for 1 hour at room temperature. Primary antibodies (listed in Table 3) were incubated overnight at 4 °C. Sections were rinsed three times in PBS and incubated with appropriate secondary antibodies diluted to 1:1000 and DAPI in blocking buffer for 1 h at room temperature. Sections were again washed three times with PBS. The primary antibodies used are listed in the following table. The following secondary antibodies were used: anti-*mouse*, anti-*rabbit*, anti-*rat*, anti-*goat*, conjugated to Alexa Fluor 488, 568 and 647 (Molecular Probes, Eugene, United States). Nuclei were stained in DAPI solution (1:2000) and slides were mounted in DAKO fluorescent mounting medium. As for the AP reaction, SIGMAFAST™ Fast Red TR was used and visualized by confocal microscopy (Leica) at 568 nm.

### 2.6. Imaging

Fluorescence microscopy images were captured under the LSM 780 confocal microscope (Carl Zeiss, Jena, Germany); transmission microscopy images were acquired with either the Olympus Ax70 or the Zeiss Axioscope 2 Plus. x. Adobe Photoshop software was used for image processing.

### 2.7. RT–PCR

Total RNA was isolated from the embryonic whisker pad using RNAeasy Mini Kit (Qiagen, Hilden, Germany) according to the manufacturer’s instructions. Total RNA was extracted, and 250 ng of each sample were reverse transcribed using the Superscript III enzyme and random primers (Life Technologies, Carlsbad, United States).

### 2.8. Quantitative PCR

For qPCR, 1 µL of cDNA was amplified with the Taqman Universal Mastermix II (Life Technologies) in a 10 µL total reaction volume; 5 µL of the Mastermix, 1.5 µL of CDNA, and 3.5µL of assay mix were included in the reaction. The primers were bought from Applied Biosciences. The TaqMan assays were performed using a 79,000 HT Fast Real-Time system (AB). For data analysis, the mouse Eef1alpha, β-actin, Gapdh, and Tbp housekeeping genes were used as internal controls. Gene expression profiling was achieved using the Comparative CT method (DDCT) of relative quantification [21] using the SDS 2.4 software (Applied Biosystems, Waltham, United States).

### 2.9. Interspecies Sequence Comparison

The comparison of the conserved noncoding elements and deletions in *mouse–rat*, *mouse–guinea pig*, *mouse–squirrel*, *mouse–rabbit*, *mouse–human*, *mouse–chimp*, *mouse–gorilla*, *mouse–orangutan*, *mouse–rhesus*, *mouse–dolphin*, *mouse–cow*, *mouse–cat*, *mouse–dog*, *mouse–horse*, *mouse–elephant* was performed using the Vertebrate Multiz Alignment and Conservation Track in the UCSC genome browser, using a window size of 2 kb. HiC data on topological associated domains (TADs) from ES cells were obtained from http://3dgenome.fsm.northwestern.edu/view.php (accessed on 11 April 2020) [22,23].

### 2.10. 4C-Seq

For each sample, we harvested at least 1 × 10^7^ cells and obtained 7–10 μg of output double-digested, double-ligated DNA. We collected 120 embryonic whisker pads, both at E12.5 and E13. NlaIII and DpnII were used as primary and secondary cutters, respectively; ligation was performed by using the Concentrated T4 DNA ligase from Promega. Primer sets for *Lef1* promoter and Leaf enhancer are described in Table 4.

PCRs were multiplexed and sequenced with Illumina HiSeq2000. Then, 4C-seq reads were processed through the HTS station according to previously described procedures [24,25] and visualized with gFeatBrowser (http://www.gfeatbrowser.com, accessed on 12 January 2022). Briefly, raw reads were demultiplexed and aligned to the *mouse* mm10 reference genome (GRCm38). The regions directly surrounding the viewpoints, chr3: 131′104′979–131′112′546 for the *Lef1* promoter viewpoint and chr3: 131′016′310–131′022′769 for the Leaf viewpoint, were excluded from the analysis. Normalization was performed by dividing the fragment scores by the mean of fragments scores falling into a region defined as +/−1 Mb around the center of the bait coordinates. Smoothed signals were obtained by applying a running mean algorithm with a window size of eleven fragments. The profile corrected data were generated by applying an approach similar to the one described in [26] using a fit with a slope −1 in a log-log scale [27]. The 4C-Seq data are deposited in GEO under the accession number GSE193356 (Table 5).

### 2.11. Statistical Analysis

To compare the contacts between the *Lef1* promoter and Leaf, in the E12.5 whisker pad and the adult kidney, we considered the normalized and profile corrected fragments from region chr3:131′008′663–131′026′430 for Leaf viewpoint, and those from region chr3:131106987–131227057 for *Lef1* promoter. Statistical differences were assessed by applying an unpaired non-parametric two-tailed Mann–Whitney test after having excluded the normality of the data with a D’Agostino and Pearson omnibus normality test. Differences were considered significant if the *p*-value, *p*, was less than 0.05 and shown in figures as * (* *p* < 0.05, ** *p* < 0.01, *** *p* < 0.001, **** *p* ≤ 0.001).

For Leaf, fragments of region Chr3:131008663–131026430 (mm10) were analyzed whereas, for the *Lef1* promoter and coding sequence, the fragments of region Chr3:131106987-131227057 were analyzed. The normality of the data was excluded by the D’Agostino and Pearson omnibus normality test. Statistical differences were assessed by applying an unpaired non-parametric two-tailed Mann–Whitney test. Differences were regarded as significant if *p* < 0.05, and significances are shown in figures as * (* *p* < 0.05, ** *p* < 0.01, *** *p* < 0.001, **** *p* ≤ 0.001). Analyses were conducted using GraphPad Prism version 8.0.

### 2.12. CUT&Tag

CUT&Tag was performed on wild-type E12 whisker pads dissociated to a single cell suspension according to the v1.5 Epycypher Protocol (https://www.epicypher.com/resources/protocols/cutana-cut-and-tag-protocol/, accessed on 12 January 2022). Rabbit anti-H3K4Me1 (Abcam, Ab8895) and IgG (Cell Signaling, 2729S) were used at 0.5 ug per sample. Samples were sequenced at a depth of 10Mio each with the Illumina NovaSeq 6000 system (51 bp paired-end runs).

The peak calling for the CUT&Tag experiments was performed per each replicate with SEACR v1.3 [28] using the norm and stringent parameters. Due to the poor distribution of fragment lengths observed for IgG replicate 1 and the subsequent low reproduction rates (see Appendix A), we rather called the peaks without IgG by setting the top fraction of peaks considered to 1%, 2%, 3%, 4%, and 5% to assess the overall behavior. The peaks represented in the figures are the ones obtained from the top 1%. CUT&Tag data are deposited in GEO under the accession number GSE192851 (Table 5).

## 3. Results

### 3.1. *Prdm1* Is an Essential Gene for Whisker Follicle Development

We localized *Prdm1* expression by anti-GFP immunofluorescence in *Prdm1*mEGFP reporter [17] embryonic whisker pads during the first stages of whisker development (Figure 1A,B).

The expression of mEGFP in heterozygous embryos can be detected in a specific cluster of mesenchymal cells underlying the monolayer of embryonic epidermis (referred to as stage 0 of whisker development, [29]) that will later form the whisker epidermal placode. It continues to be expressed in this compartment until stage 4, when its expression is turned on in the inner root sheath (IRS) of the follicle. At stage 5, mEGFP expression disappears in the DP (Figure 1C). Those results are validated by *Prdm1* immunohistochemistry (IHC) in wild-type whisker pads (Figure 1D).

On the other hand, *Prdm1* expression in the dermal condensate of both head and back pelage hair follicles starts at embryonic day E14.5 but is transient as it disappears at E16.5 (Appendix A).

To position *Prdm1* in the molecular cascade leading to whisker formation, we generated *Prdm1* knockout embryos. It was previously shown that the constitutional knockout of *Prdm1* leads to severe impairment of the placenta resulting in early embryonic lethality [16] Therefore, we used a *Sox2*Cre deleter strain to bypass the placental developmental halt [20,30]. By combining *Sox2*Cre with a floxed *Prdm1* allele [14], we were able to obtain viable E17.5 *Prdm1* homozygous conditional knockout embryos (referred to as cKO1 in this study). We started by analyzing E12.5 and E13.5 cKO1 embryos, the relevant time points being when whisker follicle formation begins. cKO1 embryos recapitulate previous findings (Appendix A) [20] and, microscopically, neither whisker placode can be detected, nor the dermal condensate formation occurs (Appendix A). On the other hand, the development of pelage hair follicles is not impaired at E15.5 (Figure 2A), as observed in [20].

To understand where *Prdm1* stands in the molecular cascade leading to whisker formation (Figure 2B), we looked with fluorescent in situ hybridization (FISH) at the expression of the genes involved in the first molecular steps of whisker follicle morphogenesis, both in the epithelial and mesenchymal compartment. β-catenin is involved in establishing the first dermal signal [31]; consistent with this, *Lef1* is expressed throughout the mesenchyme of the *mouse* vibrissa pad prior to vibrissa follicle development at E11, and initiation of vibrissa follicle development is dependent on its expression [32,33]; at E12.5, its expression is confined to the epithelial placode and underlying mesenchyme. In E12.5 homozygous cKO1 mice, *Lef1* is expressed homogeneously in the mesenchyme of the whisker pad, thus, indicating that the first signal is present, even though there is no further development, which would require the physiological restriction of *Lef1* expression in the whisker placode and underlying mesenchyme (Figure 2C).

In *Prdm1* cKO1 mice, no placode formation can be observed both macroscopically and microscopically (Figure 2 and Appendix A). In the wild-type whisker pad, there is a patterned upregulation of molecules involved in the promotion and inhibition of placodal fate. This includes *Bmp4* (Figure 2D) in the pre-follicle mesenchyme, *Bmp2* (Figure 2E) in the epithelial compartment, and *Shh* (Figure 2F) secreted by the placodal cells. However, those are clearly absent in *Prdm1* cKO1 mice (Figure 2D–F). *Gli1* is not upregulated in the pre-follicular mesenchyme (Figure 2G). We found that *Wnt10b* is diffusely expressed in the monolayer of epidermal cells in the sites of whisker formation in cKO1 and does not show a marked upregulation in placodes (as in the wild-type whisker pads) (Figure 2H). Furthermore, RT-qPCR on whisker pads at E12.5 shows a statistically significant decrease in *Edar*, involved in the promotion of placode, and *Keratin 17*, whose expression arises within the single-layered, undifferentiated ectoderm of E10.5 *mouse* embryos and giving rise, in the ensuing 48 h, to the epidermal placodes [34] (Appendix A). Altogether, these data show that in the absence of *Prdm1*, the gene cascade involved in placode formation is not activated, despite the presence of the initial *Lef1* expression in the whisker placode mesenchyme.

**Figure 2 biomedicines-10-02647-f002:**
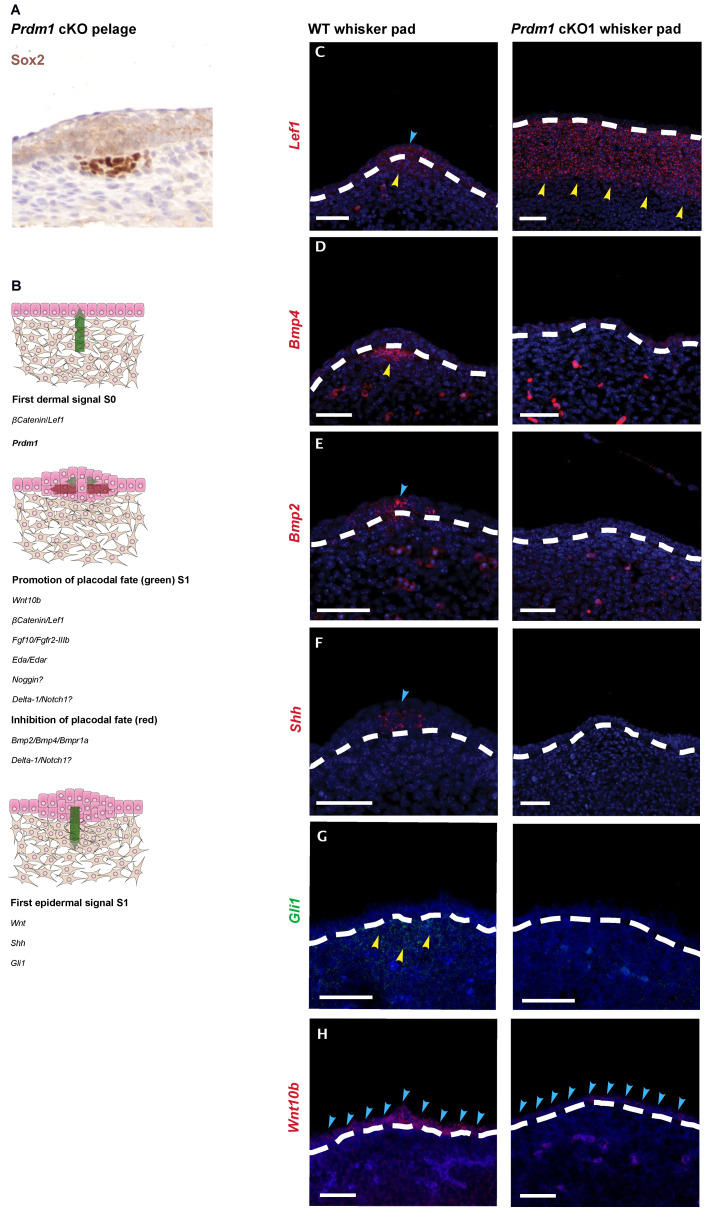
*Prdm1* conditional knockout locks the developing whisker pad at the level of the first dermal signal leaving pelage hair follicle formation unperturbed. (**A**) *Sox2* immunohistochemistry on E15.5 head skin of *Prdm1* cKO1 embryos. Pelage hair follicle placodes are present as well as the mesenchymal condensate. (**B**) Expression of early whisker developmental genes in the *Prdm1* cKO1 mouse. Molecular mechanisms underlying the early steps of whisker development. The dermis delivers a β-catenin-based homogenous first signal to the overlying epidermis in order to initiate placode formation. The placode in turns both sustains their growth and inhibits the formation of other placodes in the surrounding epidermis. The promotion of the placodal fate is sustained by several molecules including Wnt10b, βcatenin/Lef1, Fgf10/Fgfr2-IIIb, Eda/Edar, Noggin, Delta-1/Notch1, whereas the inhibition is based upon Bmp2/Bmp4/Bmpr1a and Delta-1/Notch1. Thus, the placode conveys a first epithelial signal, leading to the clustering of the mesenchymal cells underneath into the dermal condensate. This process mainly relays on Wnt and Shh signaling. “Image adapted with permission from [35]. Copyright 2002, Elsevier.” Fluorescent ISH on E12.5 wild-type embryos (S1 of whisker development) reveals that *Lef1* expression is confined to the epithelial placode and underlying mesenchyme (**C**); *Bmp4* the underlying mesenchyme I (**D**); *Bmp2* marks the epithelial placode (**E**); *Shh* is expressed by the placode and induces the condensation of the mesenchyme (**F**); *Wnt10b* expression is restricted in placodes (**G**); *Gli1* is upregulated in the pre-follicular mesenchyme. In E12.5 cKO1 embryos, *Lef1* expression is homogenous throughout the mesenchyme; *Bmp2*, *Bmp4*, *Shh,* and *Gli1* are no longer detectable, while *Wnt10b* fails to be upregulated in placode areas; whisker follicles cannot, thus, reach stage 1 of whisker development. The dashed lines indicate the dermo–epidermal junction. Hybridization is marked with arrows (the yellow arrows) indicate the hybridization of the probes in the placode, the yellow ones in the mesenchyme); (**H**) *Sox2* IHC on E15 head skin of *Prdm1* cKO1 mouse reveals that the development of head hair follicles is unperturbed by the deletion of *Prdm1*. Scale bar: 50 μm.

### 3.2. The Whisker Inducing Mesenchyme Contributes to Several Lineages of the Adult Whisker

To investigate the proliferative activity of *Prdm1* expressing cells, we administered a short pulse (2 h) of nucleotide analog ethynyldeoxyuridine (EdU) to *Prdm1*MEGFP pregnant mice carrying E12.5 embryos. We could observe that *Prdm1* expressing cells can be classified into two subpopulations (Figure 3A): quiescent—contiguous to the embryonic epithelium—and proliferative at its periphery (asterisks in Figure 3A).

We then aimed at identifying the progeny of *Prdm1* expressing cells in the whisker through lineage tracing by crossing the *Prdm1*Cre transgenic strain [17] with ROSAYFP mice [36] and collecting litters at different time points during embryonic development (Appendix A).

At E12.5, YFP-expressing cells are located in the whisker mesenchymal condensate recapitulating the GFP expression previously reported in the *Prdm1*MEGFP embryonic whisker pad. However, at E13.5, the population of YFP-positive cells expands and encompasses the area of mesenchyme surrounding the whisker hair germ. *Prdm1* can be detected in this area (except for the precursors of the DP) in whisker pad sections of both *Prdm1*MEGFP and WT mice at E12.5 and E13.5 (Figure 3B).

Given that *Prdm1*Cre is a transgenic strain that does not allow the spatiotemporal tracing of *Prdm1* expressing cells and that *Prdm1* is expressed in different territories, a *Prdm1*Cre lineage analysis does not allow the precise analysis of one given population. As we observed that Sox2 is co-expressed with *Prdm1* at E12.5 in the whisker mesenchymal condensate (Figure 3B), we consequently crossed *Sox2*CreERT2 [37] with ROSAYFP mice and injected tamoxifen in pregnant females at E12.5. Double transgenic litters were analyzed at E14.5, E17.5, and post-natal days (P) 1–3. Forty-eight hours after tamoxifen injection, YFP-positive cells were detected in the dermal condensate of the whisker hair germs (Figure 3C). As the epidermal downward growth proceeds, the mesenchymal YFP cells progressively encapsulate it (Figure 3C). When the whisker follicle reaches its final anatomical configuration, YFP is expressed by the whole DP, dermal sheath (DS), and abundantly by cells residing in the vascular sinuses (Figure 3D). More precisely, the latter are enmeshed with CD31 positive endothelial cells and display a perithelial position: both the latter and the expression of markers such as Ng2, Pdgfrβ, and Tnap indicate that they are pericytes (Figure 3D).

### 3.3. *Prdm1* Genetic Ablation Leads to the Reorganization of the Rodent Barrel Cortex

To study the impact of whisker loss on the nervous system, we looked at their innervation in the developing whisker pad of cKO1 fetuses. In the wild-type whisker pad, the afferent branches of the infraorbital nerve (labeled by p75NTR) encapsulate the whisker mesenchymal condensate without penetrating it; on the contrary, they terminate as free nerve endings in the KO counterpart (Figure 4A).

To evaluate the consequences of the impaired whisker innervation on the nervous system, we have adopted the *Wnt1*Cre as a deleter strain [38]. Wnt1 is expressed throughout the dorsal neural tube, thus, the Cre-Lox recombination occurs in all neural crest derivatives. This includes the whisker pad mesenchyme and, in particular, the mesenchymal compartment surrounding the developing whisker follicle. *Wnt1*Cre-driven *Prdm1* conditional KO mice (referred to as the cKO2) are viable and lack almost all the macro vibrissae, except for the 1–3 distal ones of the first row, as observed both macroscopically and microscopically (Appendix A). Macroscopically, pelage hair follicle seems to be absent only on the snout, even though microscopically they are present in the histological sections (Appendix A).

We retrieved the brains of both WT and cKO2 mice at postnatal day (P) 21 to account for the developmental maturation of the somatosensory system and sectioned their flattened somatosensory cortex tangentially to visualize the organization of the barrel cortex (Figure 4B). The cytochrome oxidase staining revealed that cKO2 barrel cortex undergoes a major rearrangement; the residual macro vibrissae are represented by enlarged barrels; the barrels corresponding to the micro vibrissae are, however, still present, even though their pattern is highly disorganized (Figure 4C). The deletion of macro vibrissae obtained with the *Wnt1*Cre deleter strain was confirmed using another delete strain active in the neural crest, the *Pax3*Cre mice (Figure 4D) [39].

To exclude the expression of *Prdm1* in the developing barrel cortex—and thus that the barrel cortex phenotype can be ascribed to the loss of *Prdm1* in the nervous system—we crossed *Prdm1*Cre with ROSAYFP mice and looked at YFP expression in the cerebral cortex (Figure 4D). YFP is expressed only by endothelial cells and not by the thalamocortical axons or by layer four cortical neurons, thus, excluding the aforementioned possibility.

### 3.4. *Prdm1* Knockouts Lead to Disrupted Trigeminal Nerve Wiring and Major Reorganization of the Barrel Cortex

We reasoned that *β-catenin*/*Lef1* might act upstream of *Prdm1* during whisker follicle development. To prove this, we obtained *Lef1* constitutional KO embryos by crossing heterozygous *Lef1*tm1Rug mice [30] and investigated *Prdm1* expression in the E12.5 whisker pad both at the mRNA and protein level.

At a molecular level, the quantity of *Prdm1* transcript in the *Lef1* KO embryos is lower compared to the wild-type counterpart, as demonstrated by RT-qPCR (Figure 5A). We sectioned the E12.5 whisker pads from *Lef1* KO and WT mice and localized *Prdm1* by IHC, focusing on the mesenchymal cells located under the characteristic surface elevations of the whisker pad that constitute the sites of whisker placode induction—hereafter referred to as domes. We analyzed the transverse sections of the embryonic whisker pad through (a) the primitive nasal cavity, (b) the vomeronasal organ, and (c) the tongue of five *Lef1* KO embryos. Out of five *Lef1* KOs, three domes did not express *Prdm1*. As for the remaining two embryos, we could observe *Prdm1* expression in 2/28 and 4/27 domes (Figure 5B,C).

### 3.5. Identification of Regulatory Elements Lost during Evolution in Whisker Morphogenesis

To understand if the de-regulation of *Prdm1* and/or *Lef1* might help to explain vibrissae reduction and their eventual loss in humans, we looked at their regulatory regions. As for *Lef1*, a transposable enhancer trap mapping to a particular locus (chr3:130,927,182–130,927,529 in mm10 assembly) suggests that its regulatory region is located on the centromeric side of the gene in the adjacent gene desert (TRACER LacZ expression database, SB line name 183038-emb20) [40]. The transgene expression can be observed at E11.5 in the brain, mammary glands, whisker pad, and the tip of tail, tissues/organs where *Lef1* is physiologically expressed during development (Figure 6A).

We next compared the multispecies alignment of animals with and without whiskers in the aforementioned locus, searching for putative regulatory elements. Conservation scored by PhastCons indicated the presence of an 878bp element conserved throughout the two categories of species (chr3:131,019,746–131,020,624), which, in turn, contains a 521 bp sub-region specifically absent in animals deprived of functional whiskers (*human, chimp, gorilla, gibbon, rhesus, baboon, and squirrel monkey*) mapping chr3:131,020,103–131,020,624 (mm10 assembly) (Figure 6C). Altogether, this DNA element that we called Leaf (chr3: 131,019,746–131,020,624) is located in the same topological associating domain (TAD) where *Lef1* resides (Figure 6D) [22,23].

**Figure 6 biomedicines-10-02647-f006:**
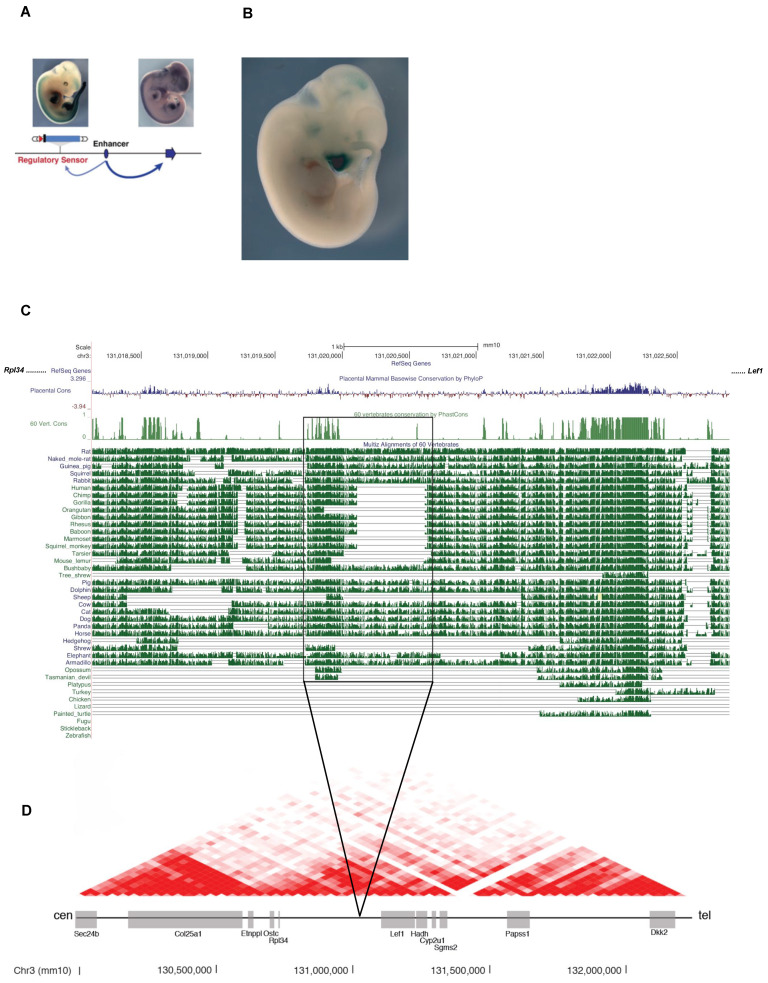
The *Lef1* regulatory region lies centromeric to its promoter and contains a primate-specific deletion named Leaf. (**A**) The TRACER database pinpoints that the regulatory region of *Lef1* resides centromeric to its promoter and that is active in E11.5 whisker pads. (**B**) A transposable enhancer trap mapping to a particular locus adjacent to *Lef1* (chr3:130,927,182–130,927,529) shows that this region is active in the whole E11.5 whisker pad (revealed by LacZ staining), coherently with the homogeneous expression of the first dermal signal. (**C**) Multispecies alignment shows a primate-specific deletion of a sequence adjacent to a conserved region. Species analyzed include animals with developed whiskers (i.e., rat, cat, dog, rabbit, guinea pig, squirrel, horse, naked mole rat, pig), primates (human, chimp, gorilla, gibbon, rhesus, baboon, squirrel monkey, orangutan, marmoset, tarsier) and the human lineages. (**D**) HiC profile centered over *Lef1* in murine ES cells [22] generated with [23] Cen: centromeric. Tel: telomeric.

We performed 4C-Seq to observe if Leaf can contact the promoter of *Lef1*. The analysis was conducted on whole micro-dissected whisker pads at E12.5 using primers positioned in the promoter of Lef1. The adult kidney was used as the negative control as *Lef1* is not expressed in this tissue (Figure 7A,B and Appendix A) [41].

We found that *Lef1* promoter scores contact mainly in cis, within an area of around 700 kb surrounding the *Lef1* locus, corresponding to its TAD. More specifically, the centromeric region contains most of the peaks of interaction, whereas telomeric contacts occur chiefly with the coding sequence of *Lef1*, extending until the end of the coding sequence of the neighboring gene *Hadh*.

In the E12.5 whisker pad, the promoter of *Lef1* contacts several centromeric regions, among which is the one containing the primate-specific deletion (Leaf). We quantified the number of contacts from the *Lef1* promoter to Leaf in the E12.5 whisker pad and compared them to the adult kidney, tissue in which *Lef1* is expressed at low levels (Figure 7B). We found high statistical significance between the scores mapping Leaf in the E12.5 whisker pad compared to the adult kidney in both normalized and profile corrected datasets (****, Mann–Whitney test, *p* = 0.0008 in normalized and *p* < 0.0001 in PC).

In the reciprocal experiment, we scored the contacts from the murine Leaf enhancer with the promoter and coding sequence of *Lef1*. Similarly, we showed a high statistical significance between the scores mapping *Lef1* in the E12.5 whisker pad compared to the adult kidney (****, Mann–Whitney test, *p* < 0.0001 in normalized and *p* < 0.0001 in PC), suggesting productive long-range interactions between Leaf and *Lef1* in the whisker pad, confirming our observations from the specular viewpoint.

To validate Leaf as an enhancer, we then performed CUT&Tag [42] on E12 wild-type whisker pads to map the presence of monomethylation of histone 3 lysine 4 (H3K4me1)—a post-translational modification enriched at active and poised enhancers. We observed a clear enrichment of H3K4me1 at the Leaf locus (Figure 7A). This result was reproducible across three wild-type whisker pads analyzed (Appendix A), which, when taken together with our previous results, clearly demonstrates Leaf as an enhancer that contacts the *Lef1* promoter (and vice versa).

### 3.6. Expression of an Ar Regulatory Enhancer Specifically Lost in Humans in the Early Steps of Whisker Development

It was recently demonstrated that a specific enhancer of the androgen receptor gene (*Ar*) is active in the murine whisker mesenchyme during development [43]. The authors proposed that the loss of this enhancer is associated with the loss of both sensory vibrissae and penile spines in the human lineage [43,44]. Notably, the castration or genetic deletion of the *Ar* in mice results in reduced growth of whisker follicles in mice without their full disappearance.

To understand if the *Ar* is involved in the early steps of whisker development together with *Prdm1* and *Lef1*, we performed both IHC and RT-qPCR on embryonic whisker pads at E12.5. The Ar IHC shows no signal either in the whisker placodes or in the mesenchyme underneath it at E12.5 and the RT-qPCR confirmed the absence of expression in the micro-dissected whisker pads at E12.5 and E13.5 (Appendix A), thus, excluding the hypothesis of its early involvement in whisker formation.

## 4. Discussion

Bringing to light the molecular mechanisms underlying vibrissae development and their evolutionary reduction—until their complete disappearance in humans—is key to understanding how the modifications in molecular circuitries led to their unique features specific to different organisms.

While previous work has shown that *Prdm1* and *Lef1* have an indispensable role in whisker formation in rodents [20,31], their precise positioning in the morphogenic molecular cascade has remained elusive. The mesenchymal structures constituting the adult whiskers are of neural crest origin [45]. In line with this, *Prdm1* expressing cells give rise to the dermal papilla, while the cells failing to incorporate into it migrate to surround the shaft of the forming follicle, giving rise to an adult progeny that has not yet been described [20]. Each adult whisker is innervated by branches of the trigeminal nerve and is represented in a one-to-one fashion in the barrel cortex. A synergy between the barrel cortex genetic programs and the physical presence of whisker follicles on the snout determines the development and somatotopic map of the barrel cortex, a puzzle to be completed with the ablation of whisker developmental genes. Previously, the deletion of a whisker-specific enhancer from the human androgen receptor in humans was described as a distinctive and characteristic feature of the human lineage [43,44]. Given that the *Ar* is not expressed during the early stages of whisker development and that *Ar* knockout mice still develop vibrissae [46], surely other regulatory elements—and most probably the ones of genes involved in the early phases of morphogenesis—are involved in the evolutionary loss of whisker follicles.

Our experiments clearly indicate that *Prdm1* operates at the level of the first dermal signal during whisker formation and that in its absence, the development of pelage hair follicles is not perturbed during the early stages of development [47]. Intriguingly, *Prdm1* is no longer expressed during human pelage hair follicle development [48], indicating that its upstream genes no longer induce its expression in hair follicles. *Lef1* (and thus the β-catenin-based first dermal signal) is indispensable to induce *Prdm1* expression, though the mechanism by which this occurs needs further investigation. We envision the cascade leading to whisker formation as a circuit where genes are activated in a specific sequence to start its formation and sustain its growth. As for the first dermal signal, *Lef1* and subsequently *Prdm1* represent the first genes of the circuit that trigger the successive phases (i.e., the formation of the placode first and then of the first epidermal signal).

We show that the population of cells expressing *Prdm1*/*Sox2* in the embryonic whisker mesenchyme gives rise to several lineages of the adult whisker, including the DP and DS, thus, representing a population of multipotent progenitors. This result explains the common inductive properties of the DP and DS, as reported by others [49]. Additionally, we show that it gives rise to pericytes residing in the whiskers’ vascular sinuses.

We demonstrate that in the absence of *Prdm1* expression, the afferent branches of the infraorbital nerve do not organize into plexi surrounding the mesenchymal condensates but terminate as free nerve endings. Axonal guidance is most probably absent because of the lack of nerve guidance molecules secreted by *Prdm1* positive cells. The *Wnt1*Cre driven *Prdm1* KO (cKO2) mice lack almost all macro vibrissae, although they retain micro vibrissae; we can, thus, describe *Prdm1* as strictly necessary for macro vibrissae development. Those conditional knockout mice have major rearrangements in the barrel cortex also affecting the representation of micro vibrissae with relevant evolutionary consequences.

King and Wilson [50] postulated that “regulatory mutations account for the major biological differences between chimp and human”. It was already elegantly demonstrated how enhancers regulate craniofacial morphology [51], highlighting the prominent role of enhancers in morphological evolution. The multispecies deletion we identified in the regulatory region of *Lef1* that is specific to several primates (including *human, chimp, gorilla, gibbon, rhesus, and baboon*) is flanking a well-conserved region among many mammals, altogether forming a regulatory element named Leaf. Using 4C-Seq analyses and H3K4me1 CUT&Tag, we have shown that Leaf is an enhancer in E12 embryonic whisker pads.

While great apes have lost macro vibrissae, although retaining the micro vibrissae on lips (a phenotype reminiscent of our *Prdm1* cKO2 mice), as well as cheeks and eyebrows, humans are the only known mammals to have lost both. In the model we envisage, whisker loss was a multi-step process that started during the divergence of the species and that relied on the loss of tissue-specific regulatory elements of several genes involved in whisker formation. More specifically, the loss of Leaf and other putative enhancers might have contributed to the downregulation of the expression of genes fundamental for whisker development in the snout of primates, thus, contributing to whisker loss together with other mechanisms that have yet to be identified.

## Figures and Tables

**Figure 1 biomedicines-10-02647-f001:**
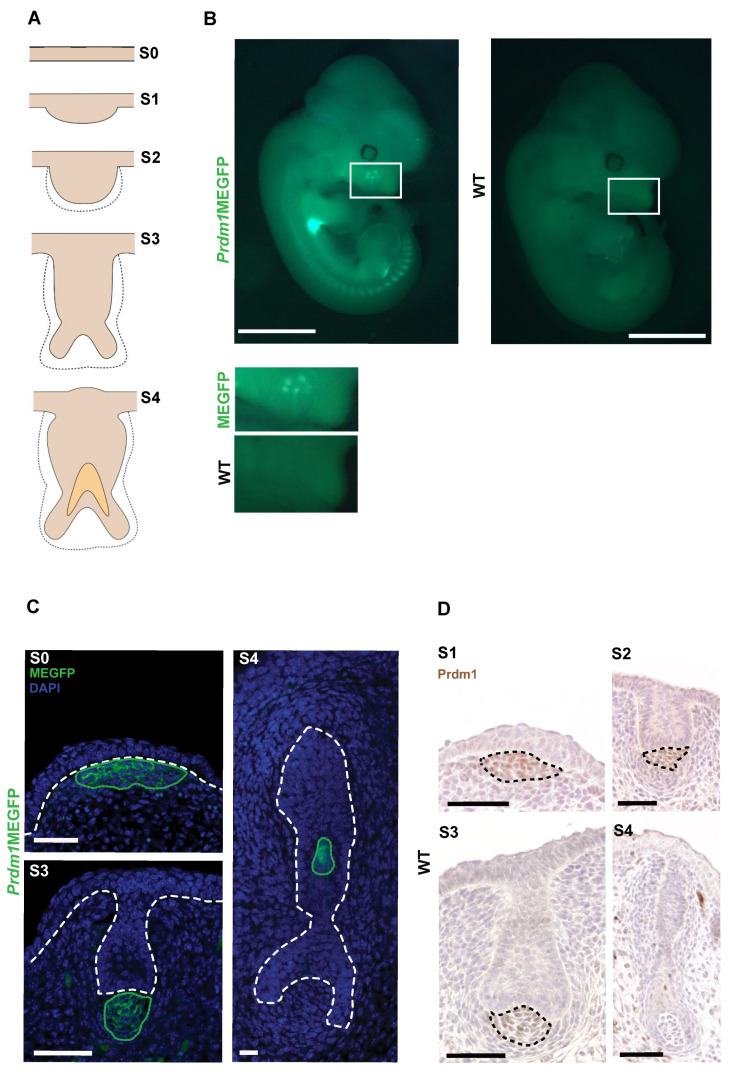
*Prdm1* is expressed in the early mesenchymal condensate during whisker development. (**A**) Schematic representation of the whisker follicle developmental stages (S1–S4). Whisker development starts around embryonic day E12.5 (day 0 of whisker development) when an early dermal condensate appears in the whisker pad below the embryonic epidermis, which will, in turn, thicken to form a placode in stage 1. Subsequently, an epidermal down growth (stage 2) and a dermal papilla (stage 3) are formed. A hollow cone (stage 4) develops by the hardening of cells belonging to the hair matrix, thus, giving rise to the inner root sheath. Image elaborated on (Hardy 1992). (**B**) *Prdm1*MEGFP versus wild-type (wt) embryo at E12.5. The fluorescent signal is observed in the developing whisker pad, forelimb, hindlimb, and somites. The whisker pads are highlighted in the white boxes. Right below full embryo pictures is the magnification of the transgenic and wild-type whisker pads. (**D**) *Prdm1* IHC on developing whisker pad (E12.5 to E15.5). Left panel, *Prdm1* is expressed in the mesenchymal compartment from stage 1 (S1) to stage 3 (S3) of whisker development. Black dashed circles envelop the areas where *Prdm1* is expressed. (**C**) GFP immunofluorescence on *Prdm1*MEGFP whisker pads. On the top left, *Prdm1* expression can be detected before placode formation (S0). The GFP expression in the dermal fibroblasts at stage 3 (S3) and in the IRS at stage 4 (S4) confirms that the reporter *mouse* recapitulates the endogenous pattern of expression of *Prdm1*. The white dashed lines indicate the epidermal–dermal junction in S0–S3 and demarcate the follicle from the surrounding mesenchyme in S4. The green dotted lines indicate the areas of GFP expression in *Prdm1*MEGFP mice. Scale bar: 50 μm.

**Figure 3 biomedicines-10-02647-f003:**
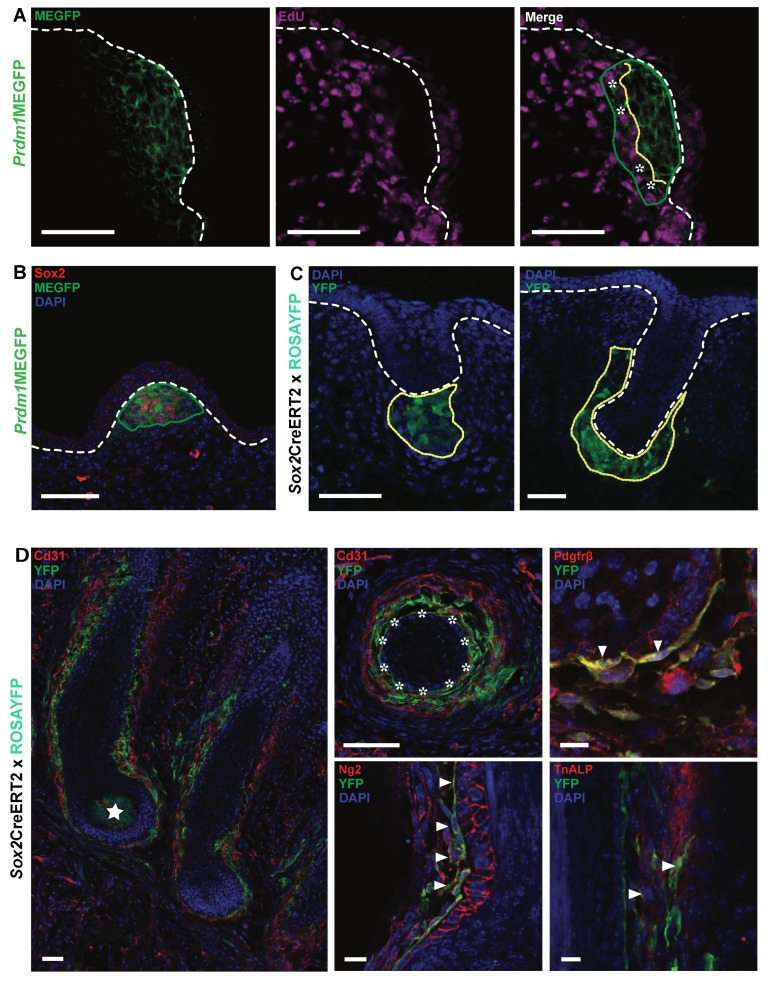
*Prdm1*/*Sox2* cells in early whisker mesenchymal condensate contribute to several lineages of the adult whisker. (**A**)The vast majority of GFP-positive cells do not incorporate I contrarily to the peripheral ones at E12.5. The yellow dashed area indicates the separation between the two populations (asterisk indicates *Prdm1*+ peripheral cycling cells). (**B**) The *Sox2* immunofluorescence on *Prdm1*MEGFP whisker pads reveals that *Sox2* marks a subpopulation of *Prdm1* positive cells. Note that Sox2 is also expressed in the putative oligodendrocytes surrounding the nerve endings surrounding the whisker pre-mesenchymal condensate. Cre expression was induced upon tamoxifen injection in E12.5 *Sox2^CreERT2^* crossed with ROSAYFP embryos and examined for *YFP* expression either in the early stages (E13, E14) or at the completion of development (E17, P3). (**C**) Note that YFP is first expressed in a cluster of mesenchymal cells right underneath the hair germ; when the latter becomes the hair peg, the YFP cluster envelops it into a mesenchymal cup. (**D**) Analysis at later time points (E17-P3) reveals the extensive contribution of YFP+ cells to several lineages of the whisker follicle including the DP (starred) and the DS (asterisks). Several YFP+ cells in close contact with the endothelial ones (expressing Cd31) can be observed inside the vascular sinuses; those cells express markers of pericytes such as Tnap, Ng2, and Pdgfrβ (white triangles indicate areas of co-expression). White dashed lines (**A**–**C**) indicate the epidermal–dermal junction. Green dotted lines indicate clusters of GFP positive cells (**A**,**B**); yellow dotted lines indicate the progeny of *Sox2* positive cells (YFP positive, C). Scale bars: 50 μm, 10 μm (Ng2), 20 μm (Tnap).

**Figure 4 biomedicines-10-02647-f004:**
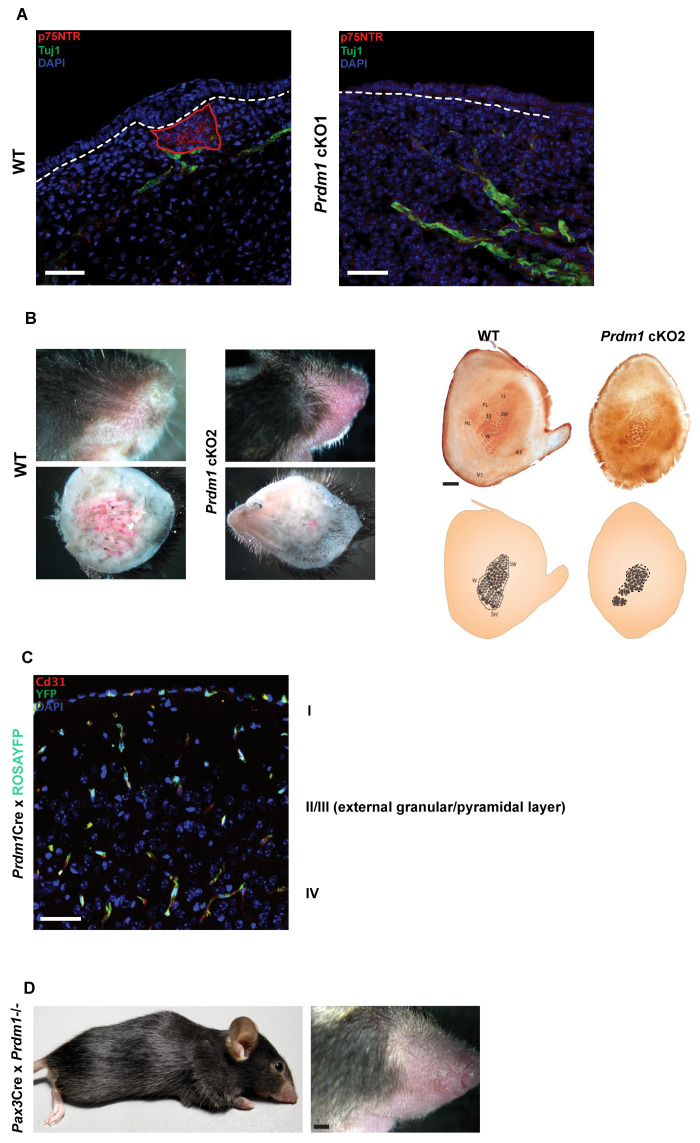
*Prdm1* knockouts lead to disrupted trigeminal nerve wiring and a major reorganization of the barrel cortex. (**A**)Tuj1 immunofluorescence on *Prdm1* cKO whisker pad at E13.5 reveals the innervation process in the early stages of whisker development. Note that the nerve fibers encapsulate the mesenchymal condensate in the wild-type, whereas they act as free nerve endings in cKO1 fetuses. The red dotted line indicates the area of dermal condensate expressing p75NTR, while the white one indicates the separation between the epidermis and the dermis. Scale bar: 50 μm. (**B**) On the left, closeup of whisker pads of WT and *Wnt1*Cre driven *Prdm1* KO (cKO2) mice. On the right, cytochrome oxidase staining on the barrel cortex of *Wnt1*Cre driven *Prdm1* KO (cKO2) mice. Note the absence of the vast majority of barrels representing the macro vibrissae and the rearrangement of the ones representing the micro vibrissae. (**C**) Cd31 (red) and GFP immunofluorescence on the somatosensory cortex of *Prdm1*Cre crossed with ROSAYFP mice. Note that the axons and neurons of layer four of the cortex have never expressed *Prdm1* and that YFP+ positive cells are of endothelial origin. Scale bar: 50 μm. (**D**) *Pax3*Cre driven *Prdm1* KO (cKO2) whisker pads recapitulate the phenotype of cKO2. Scale bar: 1 mm.

**Figure 5 biomedicines-10-02647-f005:**
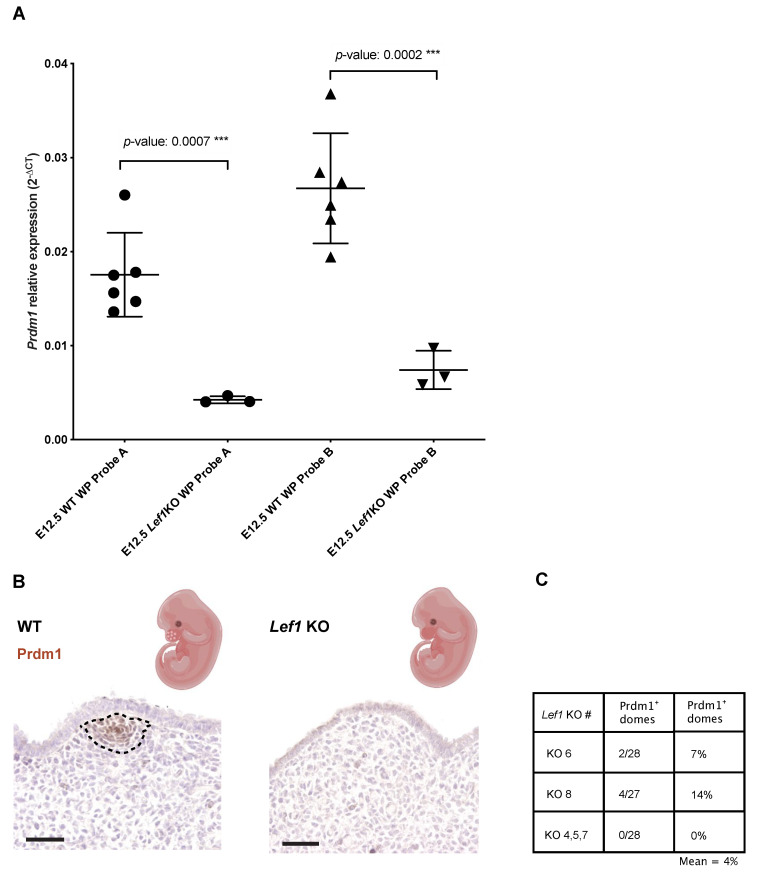
*Lef1* acts upstream of *Prdm1* during the whisker developmental program. (**A**) Quantification of *Prdm1* by RT-qPCR in both wild-type and *Lef1* KO E12.5 whisker pads (each dot represents a replicate) indicates a severe decrease in *Prdm1* expression in *Lef1* KO mice compared to the WT counterpart (*p*-value: 0.0007 ***). (**B**) *Prdm1* IHC on WT and *Lef1* KO whisker pads (E12.5). Note the absence of expression of *Prdm1* in the ectodermal elevation preconfiguring sites of whisker induction in the *Lef1* KO embryos (KO 4, 5, 7). (**C**) Quantification of *Prdm1* expressing domes in *Lef1* KO whisker pad. *Lef1* KO # indicates the number assigned to the KO embryo analyzed. ***: *p*-value ≤ 0.001.

**Figure 7 biomedicines-10-02647-f007:**
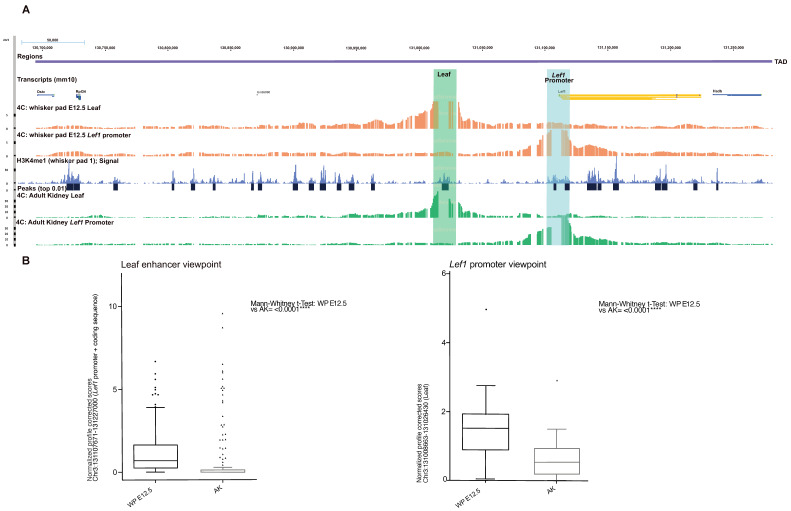
*Lef1* promoter contacts Leaf (and vice-versa) in early whisker development. (**A**) 4C-Seq on E12.5 whisker pad and adult kidney (negative control). Normalized contacts in the E12.5 whisker pad (orange) or the adult kidney (green), from the *Lef1* promoter (1^st^ track) and Leaf (2^nd^ track), in the TAD region where Leaf resides. The H3K4me1 profile on E12 whisker pads is represented together with the significant peaks (blue). The green box indicates the genomic region where Leaf resides. (**B**) Boxplots of normalized and profile corrected scores from/to *Lef1* promoter to/from Leaf region in E12.5 whisker pad and adult kidney. ****: *p*-value ≤ 0.0001.

**Table 1 biomedicines-10-02647-t001:** Mouse strains used in the study.

Strain Name	Provider
B6.Cg-Tg(*Prdm1*-cre)1Masu/J	The Jackson Laboratory
B6.129-*Prdm1*tm1Clme/J	The Jackson Laboratory
*Prdm1*MEGFP	Mitinori Saitou, MTA
B6.Cg-Gt(ROSA)26Sortm3(CAG-EYFP)Hze/J	The Jackson Laboratory
Tg(*Sox2*-cre)1Amc	The Jackson Laboratory
B6;129S-*Sox2*tm1(cre/ERT2)Hoch/J	The Jackson Laboratory
STOCK Tg(*Wnt1*-cre)11Rth Tg(*Wnt1*-GAL4)11Rth/J	The Jackson Laboratory
*Lef1*tm1Rug	Rudolf Grosschedl and Werner Held, MTA

**Table 2 biomedicines-10-02647-t002:** List of primers used for genotyping.

Primer Name	Primer Sequences
EGFP *	Primer Fw: CCTACGGCGTGCAGTGCTTCAGC
	Primer Rv: CGGCGAGCTGCACGCTGCGTCCT
Generic Cre	Primer Fw: CTAGAGCCTGTTTTGCACGTTC
	Primer Rv: GTTCGCAAGAACCTGATGGACA
*Prdm1* Cre	Primer Fw: GCCGAGGTGCGCGTCAGTAC
	Primer Rv: CTGAACATGTCCATCAGGTTCTTG
*Lef1*KotmGro	Primer 24: CCGTTTCAGTGGCACGCCCTCTCC
	Primer 25: TGTCTCTCTTTCCGTGCTAGTTC
	Primer 26: ATGGCGATGCCTGCTTGCCGAATA
ROSAYFP	Primer 1: AAGGGAGCTGCAGTGGAGTA
	Primer2: CCGAAAATCTGTGGGAAGTC
	Primer3: ACATGGTCCTGCTGGAGTTC
	Primer4: GGCATTAAAGCAGCGTATCC
*Prdm1^lox/lox^*	Primer common A: CCTGGTTAGTAGTTGAATGGGAGC
	Primer F1A: GTGCTTTTCTTGTGTTGGGAGG
	Primer F2A: AGCAGTGTTTCTGAGTGGGTGG

*: *Prdm1*MEGFP embryos were genotyped under UV light.

**Table 3 biomedicines-10-02647-t003:** List of antibodies used in the study.

Antibody	Species	Dilution	Clone	Company
Prdm1	*Mouse*	1:500	3H2-E8	Abcam
Sox2	*Mouse*	1:2000	9-9-3	Abcam
p75	*Mouse*	1:1000	9G395	UsBiologicals
GFP	*Goat*	1:1000	6673	Abcam
CD31	*Rat*	1:500	MEC13.3	BD Biopharmingen
Ng2	*Rabbit*	1:300	AB5320	Millipore
AR	*Rabbit*	1:5000	PG-21	Millipore
Tuj1	*Goat*	1:300	MMS-435P	Covance
Tuj1	*Rabbit*	1:300	MRB-435P	Covance
Pdgfrb	*Rabbit*	1:300	28E1	Cell Signaling
GFP	*Goat*	1:1000	NB100-NB1770	Novus

**Table 4 biomedicines-10-02647-t004:** List of primers used for 4C-Seq.

Viewpoint	Name	Sequence (5′->3′)
*Lef1* promoter	PromNla3Illumina	AATGATACGGCGACCACCGAACACTCTTTCCCTACACGA CGCTCTTCCGATCTTTAAACAGGGCTACCCTTAAAACCA
*Lef1* promoter	PromDpn5Illumina	CAAGCAGAAGACGGCATACGAAGGCTCAGTCTTCATCC ACACC
Leaf	EnNla2Illumina	AATGATACGGCGACCACCGAACACTCTTTCCCTACACG ACGCTCTTCCGATCTCCGGAAGCGGCTGTTCTC
Leaf	EnDpn1Illumina	CAAGCAGAAGACGGCATACGAGGTGGAGAACGGAACC CAAG

**Table 5 biomedicines-10-02647-t005:** Deposited bioinformatic data.

Data Type	Website	Accession
4C-Seq	https://www.ncbi.nlm.nih.gov/geo/query/acc.cgi?acc=GSE193356, accessed on 12 January 2022	GSE193356
CUT&Tag	https://www.ncbi.nlm.nih.gov/geo/query/acc.cgi?acc=GSE192851, accessed on 12 January 2022	GSE192851

## Data Availability

All bioinformatic data are available at GEO (NCBI) under the accession codes GSE193356 and GSE192851.

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
