# Peer review of "The Transcriptional Regulator Prdm1 Is Essential for the Early Development of the Sensory Whisker Follicle and Is Linked to the Beta-Catenin First Dermal Signal"

_biomedicines, 2022, doi:10.3390/biomedicines10102647_

Round 1

Reviewer 1 Report

In this manuscript, Manti et al. reported a study about the knockout effect of Prdm1 in mice.

The authors claim that Prdm1 is a master gene for the early development of sensory whisker follicle.

1.     First, what is a “master gene”? What is the definition? I don’t think it is mentioned in the manuscript. The evidence of Prdm1 is a “master gene” presented in the manuscript, is just that the conditional KO mice don’t develop whisker placode. Theoretically, knocking out any genes involved in whisker placode development, could potentially impair whisker placode development. Are they all “master genes”? Please provide more evidence why this gene is considered as “master”.

2.     Second, the claim that Prdm1 is wired to the Lef1 circuitry needs more evidence. The authors have already stated that “In E12.5 homozygous cKO1 mice, Lef1 is expressed homogeneously in the mesenchyme of the whisker pad, thus indicating that the first signal is intact”. It seems to me that the authors are assuming Lef1 pathway is the only pathway involved in whisker development, anything impacting whisker formation has something to do with Lef1. I think the authors need to provide more evidence to claim Lef1 and Prdm1 are in the same “circuitry”. And what exactly is a “circuitry”, do the authors mean pathway? (And why Lef1 is not considered as “master” gene in the same logic?)

3.     RNA-FISH is not a very quantifiable approach to measure gene expression levels (there is no negative control either), for example claiming Lef1 expression is intact in KO mice. qPCRs are needed.

4.     Page 19, The authors stated, “the loss of this enhancer (Ar) is associated to the loss of both sensory vibrissae and penile spines in the human lineage.” Need evidence to make this claim.

5.     Fig 5c, gene name tracks needed. And why is this specific region chosen?

6.     Fig 5d, the HiC profile centered over Lef1 is based on murine ES cells. Do the authors assume the HiC profile remain unchanged in mouse embryonic tissues?

Author Response

We thank the reviewer for the constructive suggestions and are addressing the points raised below.

We believe those helped us raise the quality of the manuscript.

The authors claim that Prdm1 is a master gene for the early development of sensory whisker follicle.

  1. First, what is a “master gene”? What is the definition? I don’t think it is mentioned in the manuscript. The evidence of Prdm1 is a “master gene” presented in the manuscript, is just that the conditional KO mice don’t develop whisker placode. Theoretically, knocking out any genes involved in whisker placode development, could potentially impair whisker placode development. Are they all “master genes”? Please provide more evidence why this gene is considered as “master”.

Prdm1 has been previously described as a master regulator in different cell types, controlling terminal differentiation of B lymphocytes (Turner et al., 1994), also governs T-lymphocyte homeostasis (Kallies et al., 2006; Martins et al.,2006), primordial germ cell (PGC) specification (Ohinata et al., 2005; Vincent et al., 2005) and stem cell maintenance in the sebaceous gland (Horsley et al., 2006) (Robertson et al., 2007).

We have defined Prdm1 as a master gene in whisker development as it is crucial specifically for whisker formation. However, being the term misleading, we decided to replace it with “essential”.

  1. Second, the claim that Prdm1 is wired to the Lef1 circuitry needs more evidence. The authors have already stated that “In E12.5 homozygous cKO1 mice, Lef1 is expressed homogeneously in the mesenchyme of the whisker pad, thus indicating that the first signal is intact”. It seems to me that the authors are assuming Lef1 pathway is the only pathway involved in whisker development, anything impacting whisker formation has something to do with Lef1. I think the authors need to provide more evidence to claim Lef1 and Prdm1 are in the same “circuitry”. And what exactly is a “circuitry”, do the authors mean pathway? (And why Lef1 is not considered as “master” gene in the same logic?)

The beta-catenin pathway has been described as the main component of the first dermal signal during appendage development (reviewed in Millar, 2002). In the chick, nuclear β-catenin is found transiently in the dense dermis underlying the feather tract 2 days before the appearance of molecular and morphologic signs of placode development (Noramly, 1999; Millar, 2002). Consistent with this, Lef1 is expressed in the mesenchyme of the mouse vibrissa pad prior to vibrissa follicle development, and initiation of vibrissa follicle development is dependent on this expression (Kratochwil et al., 1996). However, to the present moment, no other pathway has been linked to the first dermal signal to our knowledge.

Concerning the circuitry, we have envisioned the cascade leading to whisker formation as a circuit where genes are activated in a specific order to start and complete an epidermal appendage. As for the first dermal signal, Lef1 and subsequently Prdm1 represent the first genes of the circuit that trigger the successive phases (formation of the placode and formation of the first epidermal signal). However, given that we are not providing further information on the circuit, we opted to clear the term out and rephrase the title accordingly.

Concerning Lef1, while its KO arrests whisker development at the level of the first dermal signal, it also impacts the development of all epidermal appendages including pelage hair follicles, mammary glands, and teeth; thus it is not whisker specific, but it stands more as a generic dermal signal prompting epidermal appendage formation. We have opted to substitute the term master gene with essential gene to avoid any lexical and conceptual confusion among the readers.

  1. RNA-FISH is not a very quantifiable approach to measure gene expression levels (there is no negative control either), for example claiming Lef1 expression is intact in KO mice. qPCRs are needed.

Given that the cells forming the developing whisker represent the vast minority of the developing whisker pad, we have reasoned that RNA-FISH would be the most suitable technique to display the expression of the morphogens investigated and limited the use of qPCRs only to the genes where no functional probe was available. In fact, when micro-dissecting the whisker pads, the isolation of forming whiskers alone is not technically feasible and the cell pellet can contain other cell types where the morphogens are expressed, thus not allowing a quantification that is strictly representative of whisker morphogenesis; this is not true for the case of RNA-FISH, where every single area of whisker follicle formation has been scrupulously investigated under the microscope. Given that all lines are cryopreserved at the moment, the qPCRs for the genes cannot be performed.

Concerning Lef1, we apologize for the confusion. Lef1 is expressed throughout the whisker mesenchyme at the level of the first dermal signal, occurring at e11.5 in the wild-type embryos, as previously illustrated by Kratochwil (Kratochwil et al., 1996). It, later on, undergoes a restriction of expression to the placode and the underlying mesenchyme (detectable in e12.5 whisker pads). As for e12.5 cKO embryos, we could detect a broad expression of Lef1 throughout the whisker mesenchyme, recapitulating Lef1 distribution at the level of the first dermal signal in wild-type e11 mice (Kratochwil et al., 1997). However, the restriction of the signal to the placode and the underlying mesenchyme does not occur, indicating a halt in the development.

We have revised the text to make it more clear to the reader.

  1. Page 19, The authors stated, “the loss of this enhancer (Ar) is associated to the loss of both sensory vibrissae and penile spines in the human lineage.” Need evidence to make this claim.

This concept has been introduced in the literature by McLean in the article “Human-specific loss of regulatory DNA and the evolution of human-specific traits” (McLean et al., 2011). The correct reference has been added to the revised version of the article.

  1. Fig 5c, gene name tracks needed. And why is this specific region chosen?

We believe the reviewer meant figure 6c. The “gene name” track is in the figure, however, no genes are displayed as we are displaying a region between Rpl34 and Lef1, where the Lef1 regulatory region resides (as also indicated by the TRACER experiment). We have amended the figure to put those genes in perspective and facilitate the reader in comprehension (also highlighting where the deletion stands in the TAD).

  1. Fig 5d, the HiC profile centered over Lef1 is based on murine ES cells. Do the authors assume the HiC profile remains unchanged in mouse embryonic tissues?

It has been previously demonstrated that TADs are largely stable when compared between embryonic stem cells and differentiated cells (Dixon et al., 2012, 2015; Nora et al., 2012; Tena and Pereira, 2021). Despite the rearrangement at the sub-megabase scale (de Laat and Duboule, 2013) that can occur during differentiation, we expect the profile to be mostly unchanged. Furthermore, illustrating the TAD aims only at proving the information that the Lef1 regulatory region (TRACER data) and that the Leaf enhancer (UCSC track, 4C-Seq, and CUT&Tag) falls inside it.

Reviewer 2 Report

In the paper, the authors show that Prdm1 is expressed at the earliest stage of whisker development in clusters of mesenchymal cells before placode formation. Its conditional knockout in the murine soma leads to the loss of expression of Bmp2, Shh, Bmp4, Krt17, Edar, Gli1 though leaving the β-catenin driven first dermal signal intact. Furthermore, they show that Prdm1 expressing cells act as a multipotent progenitor population contributing to the several lineages of the adult whisker. Genetic ablation reverberates on the organization of nerve wiring in the mystacial pads and leads to the reorganization of the barrel cortex. Furthermore, they show that Lef1 acts upstream of Prdm1 and that the deletion of a Lef1 enhancer, named Leaf, is responsible for the disappearance of vibrissae in primates.

The study is well-thought out and well performed. The authors have added new information to the previous literature on this topic.

The paper deserve to be published upon some minor editorial corrections (for instance, lines 224-227 should be deleted…)

Author Response

We thank the reviewer for the comments and have performed the editorial corrections that have been requested.

Round 2

Reviewer 1 Report

I appreciate the authors’ effort to address my concerns. The manuscript has improved substantially. I recommend acceptance in its current form.